# Restoring Hover on Touchscreens Using a Mouse Pointer

## Abstract

Hover is a fundamental interaction mechanism in desktop environments, enabling both visual preview before action and physical hover, where users can rest a finger without triggering input. These capabilities are absent on mobile touchscreens due to the direct nature of touch input, resulting in inconsistencies between the desktop and mobile interaction phases. To address this limitation, we introduce the *tablet mouse pointer*, a novel technique that restores two-phase interaction: hover followed by tap. Five participants performed a dynamic star-rating task under three counterbalanced interaction methods—tablet mouse pointer, tablet touch, and laptop mouse—and we measured workload (NASA-TLX), usability (SUS), and quantitative performance. Results show that the tablet pointer improves selection accuracy and reduces unintended input compared to direct touch, while approaching mouse-level precision, with modest increases in temporal and physical demand. This work demonstrates that desktop-style hover can be effectively simulated on mobile devices, offering a promising path to more precise, preview-rich touchscreen interactions.

## 1 Introduction

*Hover* is a fundamental interaction state in graphical user interfaces, providing immediate feedback when a pointer rests over an element without committing an action. In desktop environments, hover enhances discoverability, supports content preview, and guides users through hierarchical structures. Tooltips, submenus, and subtle visual cues are common hover-driven affordances that improve navigation efficiency and intuitiveness. The benefits of hover are well established: is signals interactivity, visually emphasizes key elements, enables progressive disclosure to reduce clutter, enriches user engagement with dynamic visual effects, and affords precise control through cursor placement. However, hover is inherently tied to input devices with pre-contact sensing, such as the mouse and stylus. Mobile touchscreens, which integrate navigation and selection into a single gesture, lack native hover capability. This one-step interaction model simplifies execution but removes hierarchical control, reduces preview opportunities, and creates inconsistencies between desktop and mobile design paradigms. The discrepancy stems from the underlying interaction models. Desktop systems adopt a two-phase sequence of *hover then click*, while mobile touch interfaces compress these into a single *tap*. As a result, mobile UIs face limitations in replicating complex behaviors and maintaining cross-platform consistency.

To bride this gap, we introduce the *tablet mouse pointer*, a virtual pointer controlled by finger gestures that reintroduces the two-stage model on touchscreens. Inspired by the desktop mouse pointer, our approach allows users to control a cursor directly on the touchscreen, reintroducing both hover states and physical hover effects without additional hardware. The first gesture activates and moves the pointer, triggering hover effects when it overlaps with interactive elements. A subsequent gesture performs selection, effectively simulating the desktop-style *hover and click* sequence as *hover and touch*. Our empirical evaluation demonstrates that this technique not only restores hover-like

functionality on mobile devices but also improves visibility, selection accuracy, and user experience in tasks requiring fine-grained control.

In summary, our contributions are as following:

- Introduces the *tablet mouse pointer*, a purely touch-based technique that restores desktop-style hover and click on mobile screens without additional hardware.

- Presents a web-based prototype and study design that combine a dynamic star-rating task with standardized workload (NASA-TLX) and usability (SUS) measures for rigorous within-subject comparison.

- Provides empirical evidence that the tablet pointer reduces selection errors and finger occlusion while approaching mouse-level precision, offering a practical path to more precise and preview-rich mobile interactions.

## 2 Related work

The absence of native hover support on mobile touchscreens has motivated a wide range of approaches to replicate or approximate hover-like interactions. Hardware-based methods have explored proximity and infrared sensing to detect finger presence before contact. Hinckley et al.'s *Pre-Touch Sensing for Mobile Interaction* introduced capacitive field sensing that anticipates input above the display [1], and Ikeda et al. proposed a hover-based reachability technique for one-handed operation on large smartphones [2]. While effective, these approaches require specialized hardware, which limits their adoption in commodity devices. Gesture-based simulations, such as long-press and two-finger gestures, are widely used but suffer from discoverability issues, latency, and memorability challenges. Other work has focused on redesigning user interfaces to compensate for the lack of hover. For example, *LucidTouch* introduced back-of-device input to mitigate occlusion [3], while *AirPen* and mmWave-based gesture recognition explored touchless or hybrid sensing techniques [4, 5]. Although these methods can improve precision or reduce occlusion, they diverge from the desktop pointer paradigm and often incur additional visual, hardware, or cognitive overhead.

A substantial body of research underscores the value of hover feedback itself. Shimizu et al. showed that hover-based visualization significantly improved efficiency when navigating layered image content [6]. Similarly, Huang et al. analyzed cursor movement and hover behavior during web search and demonstrated that such signals reveals user attention and search intent even in the absence of clicks [7]. These findings highlight hover as a powerful mechanism for non-committal feedback that enhances navigation speed, precision, and intent inference-capabilities still largely absent from mobile interfaces.

More recent work has explored pointer-like or hybrid interaction techniques across diverse contexts. Cai et al. integrated gaze estimation with thumb swipes to extend one-handed reachability on smartphones, achieving strong performance but at the cost of camera-based sensing and calibration [8]. McDonald et al. mapped smartphone motion and gaze into a virtual pointer for VR environments, approximating desktop-like cursor interaction but requiring external tracking infrastructure [9]. Comparative studies have further examined input modality trade-offs, showing that finger, stylus, and mouse each present distinct profiles in terms of speed, precision, and workload during trajectory tracing tasks [10]. Beyond selection, cursor data have also been leveraged for implicit behavioral modeling; Liu et al. demonstrated that cursor dynamics can improve calibration accuracy in mobile eye-tracking, underscoring the richness of cursor-based signals [11].

Despite this growing body of work, few studies attempt to directly replicate the desktop pointer and hover paradigm on mobile devices using only standard touch input. Existing alternatives generally rely on proximity sensing, external hardware, or gesture surrogates. Our work addresses this gap by introducing a contact-based, freely movable pointer on mobile touchscreens and evaluating its usability and accuracy in hover-like tasks.

## 3 Method

To reintroduce the concept of *hover* into mobile interfaces, we developed the *tablet mouse pointer*, a novel technique that emulates the desktop paradigm by decoupling pointer movement from activation.

This section details the design and implementation of the tablet mouse pointer as well as the prototype environment used for user evaluation. The study comprised three phases: (i) **Pre-experiment questionnaire.** Participants completed a pre-task questionnaire assessing prior device use and interaction habits. (ii) **Main experiment.** After each interaction technique, they completed the System Usability Scale (SUS) to measure perceived usability. They also completed the NASA Task Load Index (NASA-TLX) to evaluate perceived workload, using the original wording of all items. (iii) **Post-experiment questionnaire.** Upon completing all tasks, they took part in a comparative evaluation, indicating their preferred method and providing justifications. In total, each participant completed five questionnaires.

## 3.1 Pre-experiment questionnaire

To examine participants' device usage habits and prior experience, we administered a structured pre-experiment questionnaire based on a Google Form. The survey began with an informed-consent item, clearly explaining the study purpose, voluntary nature of participation, approximate completion time (2–3 minutes), confidentiality of responses, data retention period (five years), and contact information for inquiries. Participants were then asked to enter a unique participant code, their name, gender (male, female, or prefer not to say), and age. The questionnaire next probed device-use patterns for both laptops and tablets. For each device, participants reported *usage frequency* (ranging from "several times a day" to "rarely or never") and *years of experience* (ranging from "less than six months" to "10 years or more"). These detailed demographic and device-use data were collected to characterize the participant sample, ensure informed consent, and contextualize subsequent analyses of performance and usability.

## 3.2 Main experiment

The primary task was a dynamic star-rating exercise (Figure 1), in which participants aimed to select the target star rating (0.5-5.0 stars) as accurately and fast as possible. When the experiment began, stars were highlighted as the experimenter hovered over them, and participants' selections were recorded upon clicking. This study compares three interaction methods: (1) tablet mouse, (2) tablet touch, and (3) laptop mouse. Each participant performed the star-rating task with all three methods to enable direct within-subject comparisons. To prevent learning or fatigue from affecting the results (i.e., to reduce sequence effects), the order of these methods was counterbalanced. Specifically, two participants followed the sequence: (1) tablet mouse → (2) tablet touch → (3) laptop mouse, while three participants followed: (3) laptop mouse → (2) tablet pointer → (1) tablet touch. Two surveys were administered independently to all participants after each task under the three interaction settings, using the *NASA-TLX* to assess perceived workload and the *SUS* to evaluate perceived usability, providing standardized measures of user experience.

**Interaction method details.** This study compares three interaction methods: (1) tablet mouse pointer, (2) tablet touch interaction, and (3) laptop mouse. All participants used all three methods to enable direct within-subject comparisons. The tablet mouse pointer simulates hover with a virtual on-screen pointer, allowing users to preview targets before activation. (1) **Tablet mouse pointer** enables users to control a virtual on-screen pointer on a mobile touchscreen via single-finger drag gestures. This pointer replicates desktop-style interaction and supports a two-phase model: (i) Hover phase. Users drag their finger to move the pointer. When the pointer overlaps with an interactive element (i.e., a star icon), a hover state is displayed without committing an action. (ii) Click phase. After releasing the initial drag, a second tap activates the element, mirroring a desktop click. Pointer movement is absolute and independent of finger position, similar to a laptop touchpad. This indirect mapping was deliberate: it introduces a brief learning curve but enables true hover simulation and consistent two-phase interaction. To preserve native multi-touch functions, single-finger gestures were reserved for pointer control, while two-finger gestures retained system-level actions such as scrolling and pinch-to-zoom. (2) **Tablet touch interaction.** Participants directly tapped the target star with a single finger, as is typical of smartphone or tablet usage. Selection occurred instantly upon touch, with no hover or preview state. This method relies on direct finger–screen contact and provides immediate feedback but can suffer from occlusion when precise half-star ratings are required. (3) **Laptop mouse.** A standard wired optical mouse served as the control device, offering traditional desktop pointing and clicking. Participants rested the mouse on a flat surface and used the left button

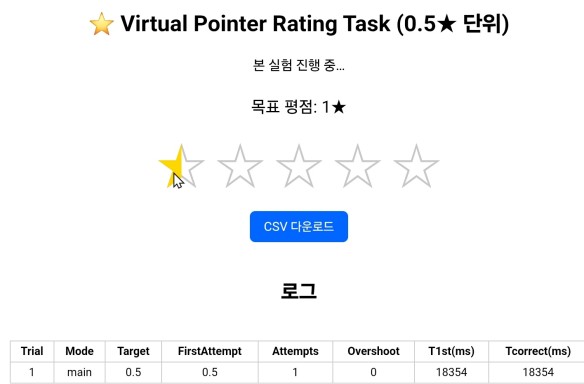

| Trial | Mode | Target | FirstAttempt | Attempts | Overshoot | T1st(ms) | Tcorrect(ms) |
|-------|------|--------|--------------|----------|-----------|----------|--------------|
| 1 | main | 0.5 | 0.5 | 1 | 0 | 18354 | 18354 |

Figure 1: **Star rating task – experiment start.** When the experiment began, stars were highlighted as the experimenter hovered over them, and participants' responses were recorded upon clicking. The graphical user interface (GUI) for the *practice mode* is shown in Figure 2 in Appendix A.

to make selections. Cursor speed and click sensitivity were set to default operating-system values to ensure consistency across sessions.

**Participants.**  Five male participants (ages 24-28) were recruited. All had prior experience with laptops and tablets, though laptops were more frequently used (ranging from multiple times daily to a few times weekly). Tablet use was more variable, from near-daily to rare, with reported experience spanning 6 months to over 10 years. All participants provided informed consent, and no personally identifying information was collected. Participants were first offered a short practice session to familiarize themselves with each interaction technique. Although not all participants used this phase extensively, it allowed acclimation to the indirect pointer mapping.

**Implementation details.**  We developed a web-based prototype in HTML, CSS, and JavaScript. Hovering previewed the target rating, and a subsequent tap confirmed the selection. The system logged hover precision, confirmation accuracy, and task completion time. Because mobile browsers block native hover events, we manually simulated hover states with custom event listeners. Tasks were implemented as local HTML files executed in Chrome. Experiments were run on a Samsung Galaxy Tab S9 (11-inch display, 2560x1600 resolution, Android 14) and a Samsung Galaxy Book Ion (15.6-inch display, 1920x1080 resolution, Windows 11) with a Logitech M110 Silent wired optical mouse. Each session lasted about 10 minutes and was conducted under consistent laboratory conditions, including controlled lighting, minimal ambient noise, and standardized seating and tablet placement. Participants received brief instructions and a short practice trial before data collection began.

**Surveys details.** These widely adopted methods—the *NASA-TLX* for perceived workload and the *SUS* for perceived usability—provided standardized measures of user experience. **National Aeronautics and Space Administration Task Load Index (NASA-TLX)** is a multidimensional assessment tool developed by NASA Ames Research Center in the 1980s to quantify the workload of aerospace workers [12]. Today, it is widely adopted in HCI research. The scale consists of six dimensions: mental demand, physical demand, temporal demand, performance, effort, and frustration. Mental demand refers to the degree of mental and cognitive effort required; physical demand refers to the physical effort exerted; and temporal demand refers to perceived time pressure. Performance reflects the level of accomplishment self-rated by participants, effort the amount of mental and physical work invested, and frustration the level of insecurity, irritation, or discouragement experienced. Each dimension is assessed on a 100-point scale, typically combined with weighting to obtain an overall score. Raw-TLX is a simplified variant that omits weighting and directly averages the six dimensions. We employed Raw-TLX to identify which specific factors contribute most to workload. **System**

**Usability Scale (SUS)** is a 10-item questionnaire proposed by John Brooke in 1986 [13]. It is widely adopted in both academia and industry for the rapid evaluation of products, services, and prototypes. Each item is rated on a 5-point scale with odd-numbered items positively worded and even-numbered items negatively worded. Scores are calculated by subtracting 1 from odd-numbered responses and computing 5 minus even-numbered responses; these values are summed and multiplied by 2.5 to yield a 100-point scale. The average SUS score is approximately 68, with scores above this indicating above-average usability and scores in the 80s considered excellent. The SUS is suitable for rapid usability assessment but not for detailed diagnostic purposes. The existing SUS questionnaire was used without modification, and the full set of items is provided in Appendix D.

### 3.3 Post-experiment questionnaire

To determine participants' comparative impressions after completing the interaction tasks, we administered a structured post-task questionnaire via Google Forms. The survey began with an informed consent item that reiterated voluntary participation, anonymity of responses, a five-year data retention period, a completion time of approximately 2-3 minutes, and contact information for inquiries. Participants were asked to enter a unique participant code, which allowed us to link their survey responses to experimental session without collecting personally identifiable information. The questionnaire then asked participants to rate the perceived similarity between the tablet pointer and the laptop mouse on a scale of 0 to 100, with 0 representing "very different" and 100 representing "very similar". Participants were also asked to report their preferred interaction method (tablet pointer or touch) and explain their choice in a free-text response. Finally, they indicated whether they encountered unexpected input errors or system issues during the tasks (e.g., "yes", "no", or "not sure"). These post-task measures were collected to assess user preferences, identify potential usability issues, and contextualize performance results within their subjective experience. Participants rated the similarity between the tablet pointer and the laptop mouse on a scale of 0 to 100, with an average score of 50 indicating moderate similarity. After completing all the questionnaires, the experimenter conducted an informal interview, asking open-ended questions about the reason for their responses.

## 4 Results

Overall, the results are presented in four parts: (i) subjective workload measured by the NASA-TLX, (ii) perceived usability assessed with the SUS, (iii) objective performance on the rating task, and (iv) post-task questionnaire responses. Together, these findings provide a comprehensive assessment of the *tablet mouse pointer* relative to tablet touch and laptop mouse interaction.

### 4.1 Pre-experiment questionnaire

The pre-experiment survey revealed that participants used laptops either several times a day or 1-3 times per week, with prior experience ranging from 5-10 years to more than 10 years. In contrast, tablet usage was more varied, with reported frequencies of 4-6 times per week, 1-3 times per week, 1-3 times per month, rarely, or never, and durations of use spanning 6-12 months, 1-3 years, 3-5 years, and 5-10 years. Overall, participants were more familiar with laptops than with tablets.

### 4.2 Main experiment

**NASA – Task Load Index (NASA-TLX).** We report NASA-TLX results in Table 1. The mouse condition consistently outperformed the other techniques across all dimensions. Compared with touch, the tablet pointer achieved higher performance and lower frustration, indicating its benefit for precise tasks. However, as expected, it required relatively greater physical, temporal, and effort demands, reflecting the additional workload involved in providing physical hover and hover-state functionality.

**System Usability Scale (SUS).** As shown in Table 2, the tablet pointer scored lower (67.50) than touch interaction, likely due to participants' limited familiarity and lack of extended training during the study. Nevertheless, the SUS results indicate that the pointer produced fewer errors than touch, albeit with longer completion times. This finding aligns with the NASA-TLX results (Table 1), where the pointer achieved higher performance but also higher temporal demand, demonstrating consistency across measures. These results also resonate with prior work such as *Hover Widgets* [14],

Table 1: **NASA-TLX results.** The NASA-TLX is a multidimensional workload assessment tool comprising six dimensions: mental demand, physical demand, temporal demand, performance, effort, and frustration. Higher scores in the performance dimension indicate better outcomes, while lower scores in the other dimensions correspond to reduced workload. Reported values represent mean $\pm$ 95% confidence interval.

| Interaction Setting | Mental | Physical | Temporal | Performance | Effort | Frustration |
|---|---|---|---|---|---|---|
| Mouse | $42.00 \pm 51.50$ | $24.00 \pm 35.77$ | $28.00 \pm 34.45$ | $95.00 \pm 10.75$ | $46.00 \pm 61.83$ | $24.00 \pm 53.12$ |
| Pointer | $44.00 \pm 43.55$ | $48.00 \pm 53.69$ | $48.00 \pm 45.11$ | $100.00 \pm 0.00$ | $56.00 \pm 55.94$ | $26.00 \pm 46.95$ |
| Touch | $44.00 \pm 42.65$ | $34.00 \pm 51.64$ | $44.00 \pm 53.12$ | $95.00 \pm 10.75$ | $50.00 \pm 57.57$ | $34.00 \pm 49.36$ |

Table 2: **SUS results.** The SUS is a 10-item questionnaire commonly used to evaluate perceived usability. Higher total scores indicate better usability: scores above 80 are considered excellent, scores around 68 represent above-average usability, and scores below 50 indicate poor usability. Reported values represent mean $\pm$ 95% confidence interval.

| Interaction Setting | SUS Score |
|---|---|
| Mouse | $81.00 \pm 23.29$ |
| Pointer | $67.50 \pm 28.02$ |
| Touch | $83.00 \pm 20.98$ |

which demonstrated that pen-based hover interactions surpass tap-only interfaces in performance and satisfaction once users pass the learning phase. Since our study did not include extended training, user preferences may have skewed toward more familiar input modes, whereas extended use could shift preferences toward the tablet pointer due to its precision and reduced error rate.

**Quantitative task performance.** Table 3 shows that the tablet pointer achieved comparable or better performance than touch interaction, with lower overshoot rates and fewer attempts. Despite participants' greater familiarity with direct touch and their lack of prior experience with the tablet pointer, the pointer consistently reduced error rates, although task completion times were longer. These findings align with SUS and NASA-TLX results, where the pointer demonstrated lower error and frustration but higher temporal demand.

Participants reported difficulty selecting half-star ratings by touch because their fingers were occluded and no preview feedback was provided. In contrast, the tablet pointer allowed users to visually assess the target rating before confirming it with a tap. This hover-tap structure increases predictability, reduces accidental selections, and mirrors the interaction logic of desktop interfaces, making it particularly useful for tasks requiring fine-grained control.

### 4.3  Post-task questionnaire

When asked about their preferred interaction method, three participants chose touch, citing its immediacy, ease of use, and familiarity with smartphone interfaces. Two participants preferred the tablet pointer, emphasizing their improved precision and reduced finger occlusion. Post-task interviews further contextualized these preferences. Participants who preferred touch stressed its

Table 3: **Quantitative task performance.** *Attempts* indicates the average number of tries required to reach the correct rating. *Overshoot* is the proportion of unintended hover confirmations beyond the target. *Time to first attempt (t1st)* is the duration from task onset to the first rating confirmation, and *Time to correct attempt (tCorrect)* is the time taken to obtain the correct rating. Lower values in all measures reflect better performance. Reported values are mean $\pm$ 95% confidence interval.

| Interaction Setting | Attempts | Overshoot | t1st (ms) | tCorrect (ms) |
|---|---|---|---|---|
| Mouse | $1.04 \pm 0.05$ | $0.01 \pm 0.02$ | $1542.46 \pm 148.28$ | $1582.80 \pm 159.39$ |
| Pointer | $1.04 \pm 0.05$ | $0.03 \pm 0.04$ | $1878.34 \pm 243.23$ | $1979.78 \pm 286.58$ |
| Touch | $1.12 \pm 0.09$ | $0.08 \pm 0.06$ | $1307.96 \pm 157.87$ | $1445.38 \pm 188.62$ |

intuitive nature, explaining that the direct mapping between input and gesture felt natural on a tablet. Conversely, participants who preferred the tablet pointer valued accuracy and visual feedback. One participant highlighted that the separation of movement and activation helped reduce accidental selections. Importantly, even some participants who preferred touch struggled to give half-star ratings, noting that finger occlusion hindered accurate targeting.

Participants also noted a structural difference between the tablet pointer and the traditional mouse. While the desktop mouse operates based on absolute positioning, the tablet pointer relies on relative finger movements, employing a touchpad-style mapping. While essential for implementation without additional hardware, this difference led some participants to perceive the pointer's behavior as inconsistent with that of a desktop environment.

Finally, regarding system errors, some participants reported experiencing unintended inputs when using touch. Despite these difficulties, several participants found the tablet pointer convenient to use, suggesting its potential value in contexts requiring fine-grained input.

# 5 Discussion

The findings of this study showed that participants had mixed experiences with the tablet pointer. While some participants found the interaction engaging and enjoyable, others preferred the touch interface due to its familiarity, especially for tasks requiring fine-grained input such as rating half stars. This aligns with previous work on pen-based devices, which demonstrated that hover interaction can increase both efficiency and user satisfaction once users pass an initial learning phase [14]. While many participants in this study chose touch because it required little or no learning effort, it is likely that preferences could shift to the tablet pointer as they become more comfortable and repeat the task over time.

A major limitation of direct touch is that it often causes occlusion and increases the likelihood of unintentional input, especially in tasks requiring precise control. The tablet pointer addresses these issues by restoring the separation between hover and touch, allowing for a clearer field of view and more intentional control. Participants unfamiliar with the technique initially made more errors, but as they focused more on the confirmation phase of the interaction, their performance improved. Interestingly, even participants who generally preferred touch pointed out that it was difficult to consistently achieve half-star ratings through direct input, underscoring the practical benefits of hover-based precision.

The tablet pointer implementation followed a touchpad-style mapping, controlling pointer movement with one-finger input and preserving system-level functionality with two-finger gestures. This design allowed participants to perceive the tablet pointer differently from a traditional mouse, highlighting that these differences were fundamental, not superficial. Overall, these results suggest that the tablet pointer offers a promising alternative to conventional touch-based interactions in mobile environments, particularly useful for tasks requiring precision not possible with direct touch alone.

# 6 Limitation and future work

Our study has several limitations. First, the small number of participants ($n = 5$) limits statistical power and the generalizability of the results. Second, the evaluation was restricted to a single task (dynamic ratings) using a single tablet device in a controlled environment, leaving open questions about applicability to other tasks and touchscreen platforms. Third, participants had limited time to adapt to the *tablet mouse pointer*, which likely biased their preferences for familiar touch input. Future work should incorporate training or longitudinal designs to understand performance after adaptation. Preferences varied depending on previous device familiarity, suggesting that a stratified study design could yield more nuanced insights. Finally, our comparisons were limited to three modes: pointer, touch, and mouse. Expanding to include stylus hover, long-press, or gesture-based techniques, and employing completely randomized task order, could reduce bias and provide a broader understanding of hover interactions.

## 7 Conclusion

We present the *tablet mouse pointer*, a novel input technique that reintroduces hover interactions to mobile touchscreens by introducing a virtual pointer that decouples movement and activation. This approach restores both the visual and physical hover states, allowing users to rest their fingers without triggering unintended inputs, bridging a long-standing gap between desktop and mobile interaction paradigms. Our empirical study demonstrates that this model supports more accurate selection, improves feedback and preview functionality, and provides intuitive and enjoyable interactions. Participants noted improved visibility and accuracy when their fingers were removed from the target. While the study was limited in scale and training duration, no adverse effects were reported and the study was conducted with informed consent. Preliminary results suggest that long-term use will lead to a shift in preference for this model, enhancing its potential to improve real-world mobile interaction design. Future work should explore pointer behavior optimization, visual occlusion reduction strategies, expanded gesture support, and longitudinal adoption studies in everyday applications.

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

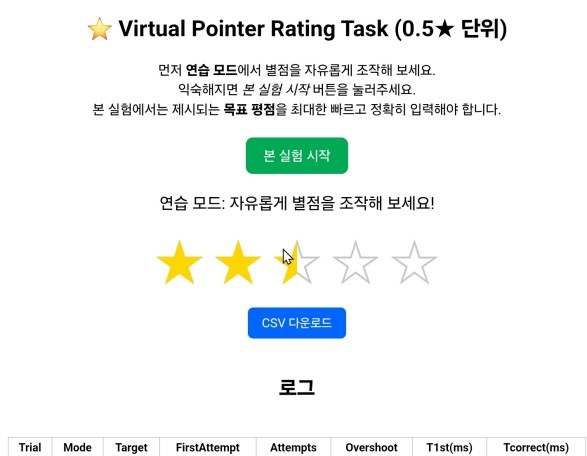

Figure 2: **Star rating task – practice mode.** Participants could freely adjust star ratings to familiarize themselves with the interaction technique before the experiment. The interface also included a logging panel and a CSV download option for recording task data.

## A  Supplementary figures

## B  Full pre-task questionnaire

1. How often do you use your laptop?
2. How long have you been using your laptop?
3. How often do you use your tablet?
4. How long have you been using your tablet?

## C  Full NASA-TLX questionnaire

1. **Mental demand:** How mentally demanding was the task?
2. **Physical demand:** How physically demanding was the task?
3. **Temporal demand:** How hurried or rushed was the pace of the task?
4. **Performance:** How successful were you in accomplish what you were asked to do?
5. **Effort:** How hard did you have to work to accomplish your level of performance?
6. **Frustration:** How insecure, discouraged, irritated, stressed, and annoyed were you?

## D  Full SUS questionnaire

1. I think that I would like to use this system frequently.
2. I found the system unnecessarily complex.
3. I thought the system was easy to use.
4. I think that I would need the support of a technical person to be able to use this system.
5. I found the various functions in this system were well integrated.
6. I thought there was too much inconsistency in this system.
7. I would imagine that most people would learn to use this system very quickly.
8. I found the system very cumbersome to use.
9. I felt very confident using the system.
10. I needed to learn a lot of things before I could get going with this system.

## E  Full post-task questionnaire

1. How similar did the tablet pointer feel to the laptop mouse pointer?
2. Which method did you prefer when performing the task?
3. Why did you prefer that method?
4. Did you encounter any unexpected input errors or system issues while performing the tasks?


