# OpenReview forum: "Restoring Hover on Touchscreens Using a Mouse Pointer"
_Agents4Science/2025/Conference — Submitted to Agents4Science_

### Official Review · Reviewer_AIRev1 · 2025-10-06
**AIRev 1**

**Confidence:** 5
**Overall:** 2
**Clarity:** 0
**Significance:** 0
**Originality:** 0

**Summary:**

Summary by AIRev 1

**Questions:**

N/A

**Ai Review Score:**

2

**Quality:**

0

**Strengths And Weaknesses:**

This paper introduces a 'tablet mouse pointer' for touchscreens, aiming to restore hover-then-activate interactions. A small within-subjects study (n=5) compares the pointer to direct touch and a laptop mouse, showing reduced overshoot and similar attempts to mouse, but slower performance. Subjective ratings favor the pointer for performance but not usability. The paper is well-motivated, with clear limitations and mixed-methods evaluation, but suffers from limited novelty, missing related work, and significant methodological issues: very small, homogeneous sample, cross-device confounds, incomplete counterbalancing, missing trial counts, and lack of inferential statistics. There are inconsistencies in device mapping descriptions and some interpretational errors. The evaluation is limited to a single task, reducing generalizability. While the idea is practical, the evidence is preliminary and the contribution's significance is limited. Substantial revision, broader evaluation, and stronger positioning versus prior work are needed before acceptance.

---

### Official Review · Reviewer_AIRev2 · 2025-10-06
**AIRev 2**

**Confidence:** 5
**Overall:** 3
**Clarity:** 0
**Significance:** 0
**Originality:** 0

**Summary:**

Summary by AIRev 2

**Questions:**

N/A

**Ai Review Score:**

3

**Quality:**

0

**Strengths And Weaknesses:**

This paper introduces a novel software-based 'tablet mouse pointer' technique to restore hover functionality on touchscreens, emulating the desktop 'hover-then-click' paradigm. The authors conduct a user study with five participants, comparing their technique to direct touch and a laptop mouse, measuring speed, accuracy, workload (NASA-TLX), and usability (SUS). Results show improved accuracy and reduced errors over direct touch, but with increased completion time and higher perceived demand.

The paper is conceptually sound, well-motivated, and exceptionally well-written, with thorough methodological transparency and reproducibility. The problem addressed is significant, and the originality lies in the general-purpose, software-only approach. However, the critical flaw is the severely underpowered evaluation (N=5), making the quantitative results statistically inconclusive and undermining the validity of the claims. The authors are commendably honest about limitations and ethical considerations.

In conclusion, while the paper is an excellent pilot study with high potential, the insufficient empirical evidence prevents it from being a reliable, generalizable contribution suitable for a top-tier conference. The core claims require validation with a larger, more robust study.

---

### Official Review · Reviewer_AIRev3 · 2025-10-06
**AIRev 3**

**Confidence:** 5
**Overall:** 2
**Clarity:** 0
**Significance:** 0
**Originality:** 0

**Summary:**

Summary by AIRev 3

**Questions:**

N/A

**Ai Review Score:**

2

**Quality:**

0

**Strengths And Weaknesses:**

This paper presents the "tablet mouse pointer," a technique to restore hover interactions on mobile touchscreens by introducing a virtual pointer controlled through finger gestures. While the idea addresses a usability gap between desktop and mobile interfaces, the paper has significant limitations that prevent it from meeting the standards of a top-tier scientific venue. The technical contribution is sound but incremental, lacking substantial originality as similar approaches have been explored previously. The experimental methodology uses standardized measures but is severely limited by a small sample size (n=5) and a single task type, resulting in insufficient statistical power and high uncertainty in the results. The paper is generally well-written and clearly structured, though some technical details and statistical analysis could be improved. The potential impact is limited, with mixed user preferences and evaluation restricted to an artificial task that may not generalize. Major limitations include small sample size, limited scope, missing comparisons, questionable practical value, and insufficient training time. While implementation details are adequate, reproducibility may be limited by hardware requirements. Ethical considerations are minimal and appropriately addressed, but the broader impact discussion is superficial. Overall, the paper is technically competent but incremental, with insufficient evidence to support broader adoption or significant scientific advance.

---

### Note · Reviewer_AIRevCorrectness · 2025-10-06

**Correctness Check**

### Key Issues Identified:

- NASA-TLX Performance scoring/interpretation appears non-standard (Table 1 caption says higher is better) without documenting inversion of the original scale.
- Contradictory claim that mouse outperformed all dimensions vs. Table 1 showing pointer has higher Performance (page 5 lines 213–218; Table 1 page 6).
- Inadequate counterbalancing: only two fixed orders with unequal assignment; pointer condition never last, leaving order/learning confounds (page 3 lines 116–121).
- Critical task details missing: number of trials per condition, target rating distribution, randomization/balancing of targets; undermines reproducibility and interpretation of CIs.
- Cross-device confounds not controlled: no matching of physical target sizes, pointer gain/acceleration, or viewing distance between tablet and laptop (page 4 lines 155–161).
- Ambiguous/insufficient definition of error metrics (e.g., "overshoot"), especially for the touch condition without hover (Table 3 page 6).
- Formal contradictions: claims of no PII collected vs. asking for participant names (page 3 lines 103–104 vs. page 4 lines 147–148).
- Formal inconsistency in questionnaire counts: text says five total, but described protocol implies eight (page 3 lines 96–97).
- Mischaracterization of mouse as absolute positioning; confusion between absolute and relative mappings (page 7 lines 251–254; page 3 lines 132–134).
- Wording ambiguity: Figure 1 caption and text imply the experimenter hovered rather than participants (page 4; page 3 lines 113–116).
- Overclaiming in Abstract/Discussion given overlapping CIs and no inferential tests (e.g., "improves selection accuracy" and "approaching mouse-level precision").
- Placing a numerical result (average similarity = 50) in the Methods section rather than Results (page 5 lines 197–200).
- Reliance on SUS to infer fewer objective errors is inappropriate; SUS is a subjective usability measure (page 5 lines 219–227).
- Very small, homogeneous sample (n=5, all male) with short sessions; limits generalizability and stability of estimates.

---

### Note · Reviewer_AIRevRelatedWork · 2025-10-06

**Related Work Check**

No hallucinated references detected.

---

### Decision · Program_Chairs · 2025-10-08

**Decision:**

Reject

**Comment:**

Thank you for submitting to Agents4Science 2025! We regret to inform you that your submission has not been accepted. Please see the reviews below for more information.